# Transformation of an Oligotrophic Sphagnum Bog during the Process of Rewetting

**Tamara Ponomareva \*, Svetlana Selyanina, Anastasia Shtang, Ivan Zubov**  **and Olga Yarygina**

N. Laverov Federal Center for Integrated Arctic Research of the Ural Branch of the Russian Academy of Sciences, 23 Severnaya Dvina Emb., 163000 Arkhangelsk, Russia; gumin@fciarctic.ru (S.S.); a_shtang@fciarctic.ru (A.S.); zubov.ivan@fciarctic.ru (I.Z.); olga.yarigina@fciarctic.ru (O.Y.)
\* Correspondence: ponomtamara@fciarctic.ru; Tel.: +7-952-259-11-01

**Abstract:** The vast peatlands of the European North of Russia were drained in the 20th century. Some of the drained areas were left without management and maintenance, which led to re-waterlogging. The current trend towards peatlands restoration requires an understanding of all the changes that have taken place in such ecosystems. The study aims to assess the changes in vegetation cover relative to changes in peat deposits in the rewetted oligotrophic bogs. The objects of research were located on the south-White Sea oligotrophic bogs. The studies were carried out using generally accepted geobotanical and geoecological methods in conjunction with the authors' method for studying the group chemical composition of peat organic matter. The species diversity, structure and spatial distribution of the vegetation cover, the structure and composition of the peat, as well as the composition of the peat organic matter have been studied. It was shown that the transformation of an oligotrophic bog during the process of rewetting manifests itself in a significant change in the vegetation species diversity, somewhat reversible concerning ecologically tolerant species. Changes occurring in the peat deposit are irreversible. That limits the possibility of restoration of species of oligotrophic habitats to the initial state.

**Keywords:** bog vegetation; physico-chemical characteristics of peat; structure of the vegetation; species diversity; oligotrophic peat





## 1. Introduction

Peatbogs and peatlands occupy about 35% of the territory of the Russian Federation [1,2]. This category of land has long been considered unproductive from the point of view of the national economy. Drainage was widely used to increase the economic efficiency of waterlogged areas in the middle of the twentieth century. It was followed by the use of drained land in agriculture, for growing commercial forests or peat extraction [1]. Long-term studies of the objects of drainage have shown that the effectiveness of these measures is not the same in different types of habitats. Oligotrophic raised bogs were recognized as the least responsive to drainage [3,4]. The forest growth effect on the raised bogs was not obvious, because of the lack of mineral nutrition and the peculiarities of the physical, physicochemical and chemical properties of the drained peat deposit. The same factors have limited the use of land in agriculture. For industry, especially for power engineering, oligotrophic peat was not of particular interest due to its low calorific capacity and low bulk density [5,6].

At the same time, a significant amount of bogs of the European North of Russia were drained from the middle to the end of the 20th century. However, economic problems at the end of the 20th century left vast drained areas without proper management and maintenance. Nowadays, drainage systems in these areas are in a diverse state. The drainage system is fully functioning on some of the objects; the ditches are destroyed or overgrown on the other and the processes of secondary waterlogging have begun. Secondary waterlogging causes a partial or complete restoration of bog phytocenoses. But

it is not clear whether the physical and chemical properties of a peat deposit are restored to an initial condition.

Nowadays, interest in bogs has shifted in favour of keeping these landscapes intact. Recent research shows that bogs provide the most valuable set of ecosystem services in a waterlogged state. The main direction of current research on wetlands is the restoration of drained areas to their initial state to neutralize the negative consequences of the long-term anthropogenic impact on bogs [7–9].

A balanced approach to the restoration of bogs requires an understanding of the entire set of changes that have occurred in this complex bio-inert system during the period of its being in a drained state. Drainage leads to significant changes in various parameters of bog ecosystems in general and their individual components in particular [10–12]. Changes under the drainage can be reversible or irreversible. In this case, the system overcomes stress and returns to its initial equilibrium state, or an external influence leads to significant transformations, up to a change in the direction of ecosystem development [13].

The vegetation cover appears to be an accessible and informative indicator of complex changes occurring both in the process of drainage and in the restoration of disturbed bogs as an indicator of the effectiveness of measures taken [14–16]. The response of the vegetation cover to external impact is cumulative and slow, and changes in the vegetation cover do not always unambiguously characterize the processes taking place in the bog ecosystem. At the same time, the vegetation cover is inextricably linked with the edaphotope (a peat deposit). Therefore, it is necessary to consider the features of the vegetation cover in conjunction with the changes occurring in the peat deposit.

Drainage and heavy use of peatlands accelerate the mineralization of organic matter [11,17], resulting in the emission of greenhouse gases (e.g., $CO_2$). Although only 15% of the world's peatlands have been drained (0.3% of the world's land cover) and are used for agriculture, livestock or forestry, peat extraction and bioenergy plantations, these drained peatlands are the source of 5% of anthropogenic greenhouse gases emissions [18].

Drainage of peatlands changes the physicochemical properties of peat soils, hydrological processes, and the chemical composition of water [11,19–21] and vegetation patterns [14]. In particular, the composition of organic matter is closely related to the species composition of peat-forming plants, with the redox and hydrothermal regimes of the deposit, and, besides, depends on climatic and hydrological factors [22,23]. Therefore, a change in the group chemical composition of peat organic matter seems to be informative for assessing the direction and depth of processes occurring in the deposit, as well as changes in the ecosystem as a whole.

The study aim was to investigate the consequences of secondary waterlogging of the drained oligotrophic bogs of the northern taiga. For this research, study sites were selected in the rewetted area and in the undisturbed and drained areas for comparison. We studied 7 test sites for the vegetation cover (vascular plant and mosses), tree rings width, peat physical (ash content, humification degree, bulk density and botanical composition) and a group chemical composition (lipids, water-soluble compounds, humic acids, fulvic acids, easily-hydrolysable compounds, cellulose, non-hydrolysable residue).

## 2. Materials and Methods

### 2.1. The Study Area

The study area was an immense oligotrophic bog system (with an area of more than 80 km$^2$) near the city of Arkhangelsk (Arkhangelsk region, Russia). The bog system is located in the northern taiga, in the highly waterlogged watershed of three rivers of the Northern Dvina basin: Brusovitsa, Shukhta and Babya. The studied bog complex is situated 55 km from the Dvina Bay of the White Sea (Figure 1). The climate of the study area is moderately cold, slightly continental Atlantic-Arctic with a pronounced influence of the White and Barents Seas. The mean annual air temperature (MAAT) of the studied area is +1.3 °C. The average air temperature in January is −12.7 °C; in July it is +16.3 °C. The growing season is usually 80–100 days with the sum of active temperatures of 1400 °C. The

Mean annual precipitation (MAP) in this area is 606 mm/year; the maximum precipitation is in July–August—70-73 mm [24].

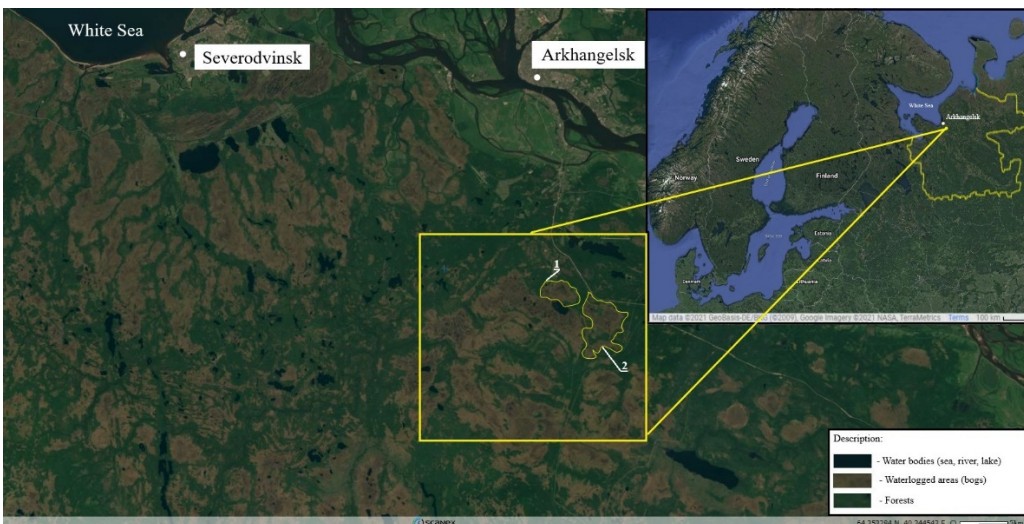

**Figure 1.** The study area in the waterlogged Northern Dvina River basin. 1—Ilas oligotrophic bog (IOB): test site UD-1 (64°18′54″ N; 40°41′14″ E); 2—the oligotrophic bog (OB-2): UD-2 (64°18′55.3″ N; 40°41′15.6″ E), UD-3 (64°19′09.7″ N; 40°40′01.0″ E), D-1 and D-2 (64°19′16″ N; 40°41′01″ E), as well as RW-1 and RW-2 (64°19′22″ N; 40°40′29″ E), modified from [25,26].

*2.2. Study Sites*

Study sites located on two oligotrophic sphagnum bogs within the selected bog system. Both bogs are typical representatives of the European suboceanic sphagnum raised bogs of the boreal zone of Russia. These are raised bogs with hummock–hollow-lake complexes in the central part. An oligotrophic type of vegetation prevails on hummocks and in hollows throughout the entire bog. The bedrocks in the study area are moraines of light and medium granulometric texture.

Test site UD-1 (64°18′54″ N; 40°41′14″ E) is located on the central plateau of the Ilas oligotrophic bog (IOB) in the southwest of the macro landscape. The Ilas bog is in a pristine state. The following test sites were established: in the central part of the plateau on the undisturbed part of the oligotrophic bog (OB-2)—UD-2 (64°18′55.3″ N; 40°41′15.6″ E), and on the slope of the plateau—UD-3 (64°19′09.7″ N; 40°40′01.0″ E). Test sites D-1 and D-2 (64°19′16″ N; 40°41′01″ E), as well as RW-1 and RW-2 (64°19′22″ N; 40°40′29″ E), were located at the edges of the OB-2 bog. The drainage system was laid out on the OB-2 bog in 1972–1974 and was intended to divert excess moisture from the roadbed during the construction of the federal road M8—"Kholmogory". The drainage system was cut by open drainage with an average distance between trenches of 100 m. An overall view of the test sites is shown in Figure 2.

*2.3. Fieldwork and Sampling*

The forest stand was investigated according to the methods adopted in Russian forestry [27,28]. At 20 × 20 m plots, the canopy tree species, coverage, forest stand composition and properties (average age, height and diameter) were assessed. The understory and shrub layer species were not assessed due to a lack of it. The basic assessments of the ground vegetation layer were made at 1 × 1 m plots with indication of broad groups of herbs, dwarf shrubs, mosses and lichens. Radial core samples were taken with a Presler age drill (Haglof, Finland) 50 cm long, with a 5 mm core diameter. The measurement of the width of the annual rings was carried out using a stereoscopic binocular microscope MBS-10 according to [29].

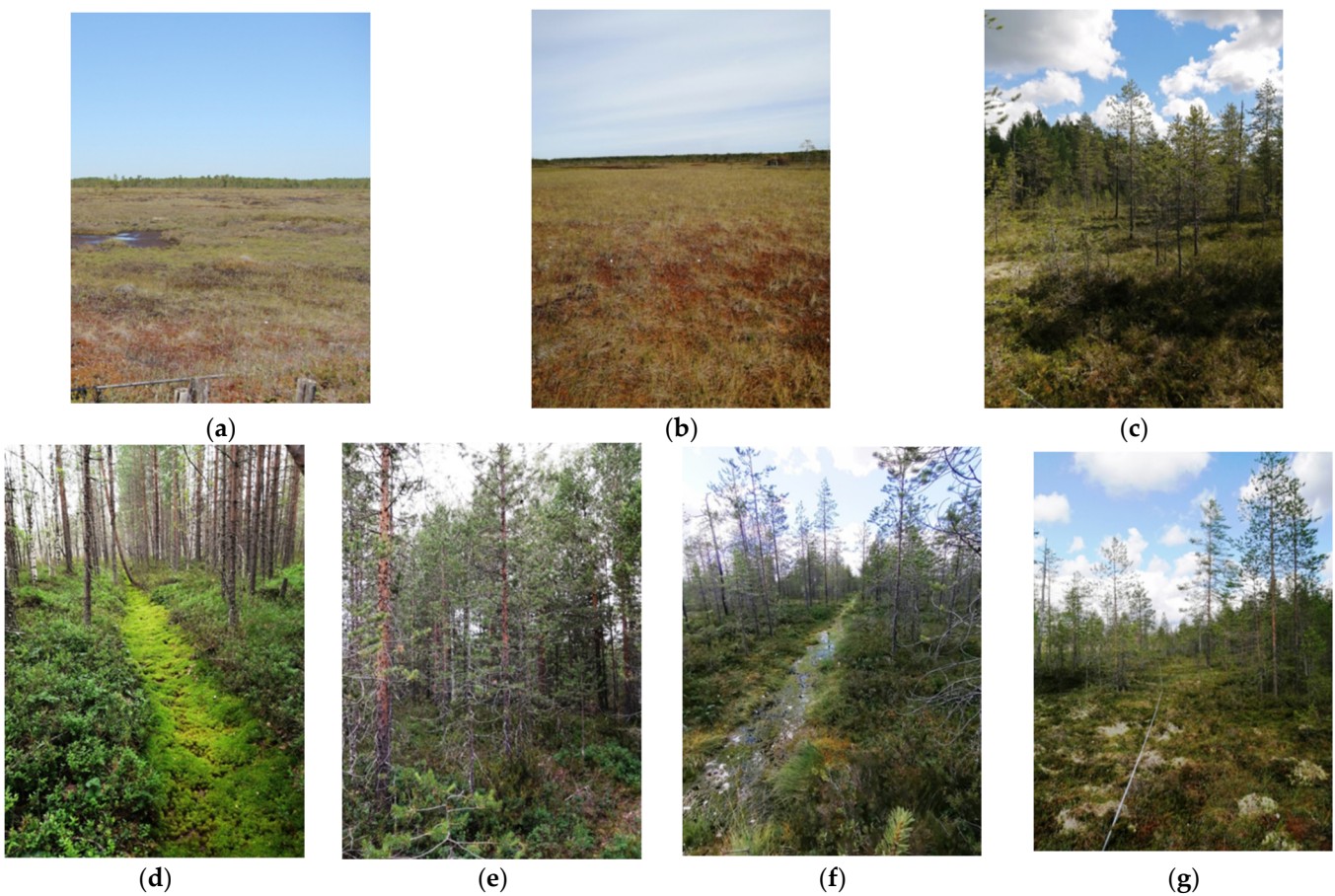

**Figure 2.** An overall view of the test sites: (**a**) UD-1; (**b**) UD-2; (**c**) UD-3; (**d**) D-1; (**e**) D-2; (**f**) RW-1; (**g**) RW-2.

The study of the vegetation cover was carried out by laying out a transect 50 m long. At undisturbed study sites, transects were laid out in the most characteristic place of the bog plant communities (4 transects). At the drained and rewetted study sites, transects were laid out in the center of the inter-trench space and along the drainage ditch (2 transects per study site). The geobotanical description of the test sites was carried out according to the methodological recommendations and standard techniques at the plots $0.5 \times 0.5$ m [30–34]. The spatial distribution of species was documented in detail by measuring their frequency of occurrence. Determination of the abundance was carried out using an established procedure by measuring the projective cover of enlarged vegetation groups, i.e., herbaceous, dwarf shrubs, sphagnum mosses, green mosses and lichens [31,32,35]. The species diversity of the herbaceous, dwarf shrub and lichen layers was determined according to the field guide [36]. The field guides [37,38] were used to identify the representatives of the moss layer plant communities

Representative peat samples were taken by layer-by-layer drilling using a stainless steel core sampler for peat deposits P 04.09 (EIJKELKAMP, Netherlands) according to [39]. Oligotrophic peat samples taken for laboratory studies were dried to an air-dry state and sieved through a sieve with a mesh size of 2 mm. The degree of peat decomposition was determined in the field by a visual method according to Von Post [40] and Tyuremnov [41].

*2.4. Laboratory Analysis*

Plant residues in the peat were identified according to [42] using an Altami Bio 2 laboratory microscope complete with a Ucmos 03100KPA digital camera and Altami Studio software. Plant remains were divided into groups: sphagnum mosses, other mosses, herbaceous remains and woody remains.

The moisture content of air-dry peat samples was determined by oven drying of peat with temperature $105 \pm 5$ °C according to [43]. The ash content was determined by the method of loss on ignition (LOI) of peat samples (1–3.0 g) in the muffle chamber at 800 °C for 3 h [43]. The bulk density of peat was determined in laboratory conditions for air-dry samples (fraction 0–2 mm) following [44]. The method is based on weighing a sample of known volume (sampler volume was 100 mL).

The group chemical composition of the organic matter of the peat samples was carried out according to the certified method [45]. The assessment of the group chemical composition of peat samples was carried out by sequential disassembly of its polymer matrix using solvents of different nature. The components of the organic matter were sequentially isolated from the initial peat sample: lipids by treatment with ethoxyethane in a Soxhlet apparatus; biopolymers of a humic nature by treatment with 0.1 N·NaOH; easily hydrolysable compounds by treatment with 5% HCl; then, the content of Klason lignin (by treatment with 72% $H_2SO_4$) and cellulose were determined in the residue. The content of group components was calculated by the gravimetric method.

### 2.5. Statistical Analysis

The primary results of the study were summarized using the data analysis package (Descriptive statistics) of the MS Excel program. To determine the significance of differences between the obtained independent samples, an analysis of variance using the Kruskal–Wallis H-test and the Mann–Whitney U-test was used. The use of nonparametric analysis is since the obtained values do not satisfy the requirements of a Gaussian distribution [46]. Nonparametric statistical analysis of independent samples was performed using the SPSS Statistics 11 software.

## 3. Results

### 3.1. Vegetation

#### 3.1.1. Tree Layer

The undisturbed test sites are oligotrophic dwarf shrub-sphagnum phytocenoses on oligotrophic peats of different thickness. An overall view of the test sites is shown in Figure 3. The drained test sites at the present stage are dwarf-sphagnum pine forests on peat soils. In pristine areas of studied bogs, the natural forest stand is confined to the hummock of the hummock–hollow complex. Therefore, the forest stand was investigated on the hummocks adjacent to the investigated carpet plant communities. The characteristics of the tree layer of the investigated test sites are presented in Table 1.

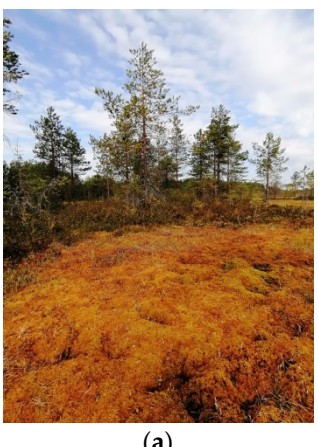 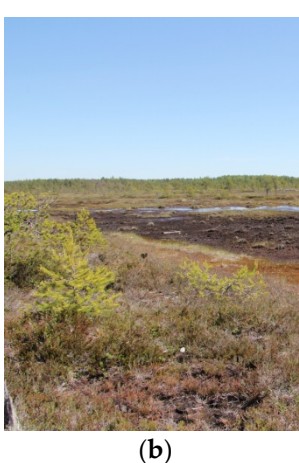 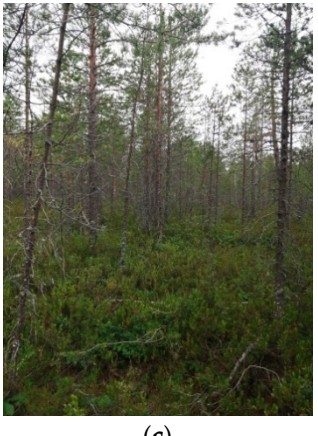 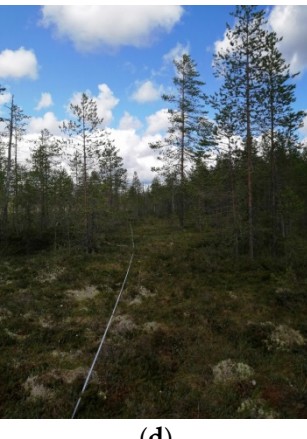

(**a**)  (**b**)  (**c**)  (**d**)

**Figure 3.** The vegetation cover of test sites: undrained test sites (**a**) UD-1; (**b**) UD-2; (**c**) effectively drained test site D-2; (**d**) rewetted test site RW-2.

**Table 1.** Properties of the forest stand of test sites.

| Test Site | Forest Stand Composition | Species | Average | | | Relative Crown Density |
|---|---|---|---|---|---|---|
| | | | Age (Years) | Height (m) | Diameter (cm) | |
| D-1 | 70% pine 30% birch | pine | 55–110 | 6.8 | 5.9 | 0.5–0.6 |
| | | birch | - | 7.0 | 5.0 | |
| D-2 | 100% pine | pine | 50–123 | 8.5 | 7.4 | 0.6–0.7 |
| RW-1 | 100% pine | pine | 49–99 | 4.8 | 5.1 | 0.2–0.3 |
| RW-2 | 100% pine | pine | 70–100 | 4.3 | 4.5 | 0.2–0.3 |
| UD-1 | 100% pine | pine | 55–235 | 2.4 | 4.3 | 0.2–0.3 |
| UD-2 | 90% pine 10% birch | pine | 50–135 | 2.7 | 4.1 | 0.2–0.3 |
| | | birch | - | 3.6 | 2.7 | |
| UD-3 | 100% pine | pine | 38–92 | 3.8 | 5.6 | 0.2–0.3 |

The drainage of the studied sites was carried out in 1972–1974. At that time, the natural forest stand was classified as young and middle-aged stands. At the moment, the natural forest stands of the drained sites are middle-aged and mature. The pine stand on the undrained study sites is very uneven-aged (38–235 years). The average height and diameter are the lowest on undrained study sites (2.4–3.8 cm and 2.7–4.1 cm, respectively). The drainage has a weak but positive effect on the tree layer. On the drained and rewetted study sites, the average height is higher (6.8–8.5 m and 4.3–4.8 m, respectively) than on pristine bog (2.4–3.8m). The average diameter is also higher on the drained study sites (4.5–7.4 cm) than on the undrained study site (2.7–4.1 cm). Although the test sites D-1,2 and RW-1,2 were drained simultaneously, the average values of tree stand are higher on the drained study site. The crown density of pine stand on the drained areas also is much higher than on rewetted or undrained study sites (0.5–0.7 relative to 0.2–0.3). At the same time, the site class of the forest did not increase.

The dynamics of annual radial growth was investigated by measuring the width of annual rings of model pine trees (the trees had values of the height and diameter close to the average values of the whole studied forest stand) (Figure 4). The response of the forest stand to drainage is manifested after 4–6 years. In this case, the average width of the annual ring increases to 0.8–1.5 mm relative to the undisturbed area (0.2–0.3 mm). On the drained study area there is a stable increase of the radial growth relative to the undisturbed study area (up to 0.8–1.0 mm) (Figure 4). On the rewetted study area, the annual radial growth gradually decreases (starting from 1980, according to Figure 4). Its active decrease is observed after 1997–1998 to the values lower than in the undisturbed area.

3.1.2. Ground Vegetation Cover

To assess the state of plant communities under conditions of different hydrological regimes, we studied the structure and spatial variability of the ground vegetation cover on the study sites using the transect method. At the same time, data were obtained on the species diversity and abundance of plant groups (herbaceous plants, shrubs, sphagnum mosses, green mosses, lichens), and a dominant species was identified in each group of plants. The similarity of the ground vegetation cover on test sites with similar hydrological conditions was confirmed statistically (the Mann–Whitney U-criteria are higher than the standard value ($\alpha = 0.002$), and the Kruskal–Wallis H-criteria are higher than the standard value ($\alpha = 0.05$)). Nonparametric analysis of variance showed that the differences between the study sites are statistically significant, according to [46].

The results were averaged over test sites: (1) UD-1, UD-2 and UD-3; (2) D-1 and D-2; (3) RW-1 and RW-2. The standard deviation does not exceed 15–20%. The species diversity and occurrence of certain types of vegetation are shown in Figure 5.

The average projective cover of vegetation groups in the study sites is shown in Figure 6.

## 3.2. Peat Deposit Characteristics

### 3.2.1. Degree of Decomposition, Botanical Composition and Bulk Density of the Peat

The undisturbed study site (UD-1) is characterized by a thick peat deposit. The average thickness of the deposit is 230–275 cm. On the central plateau, it can reach 600–800 cm. The peat deposit has a homogeneous botanical composition. It is composed of sphagnum mosses with a small admixture of cotton grass and shrubs, as well as pinewood in the bottom layers (225–275 cm). The degree of decomposition, estimated in the field, is R = 5–10% in the upper part of the deposit with a determinate increase with the depth to R = 20–25%. The groundwater level during the entire research was 0–5 cm, even during the summer low-water period.

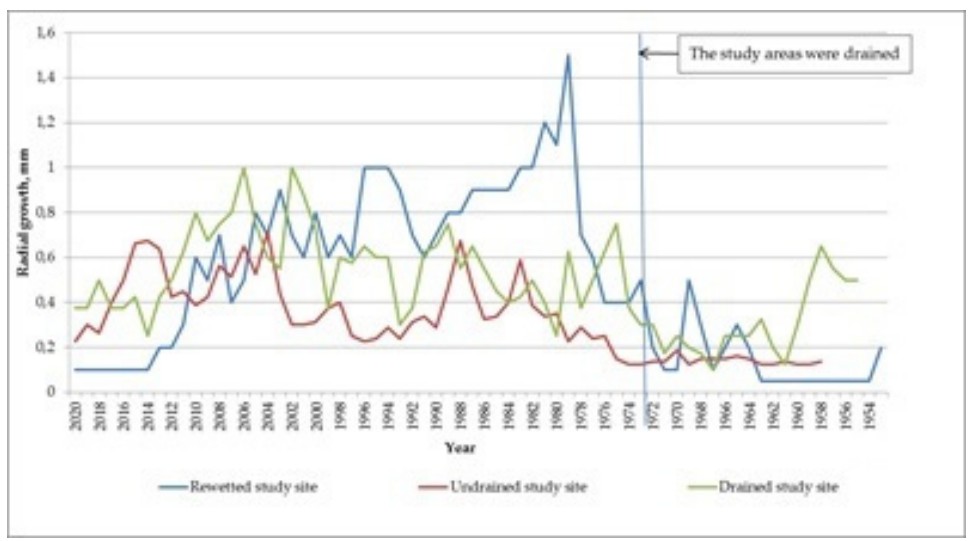

**Figure 4.** Dynamics of the absolute values of the annual radial growth of pine on the study sites.

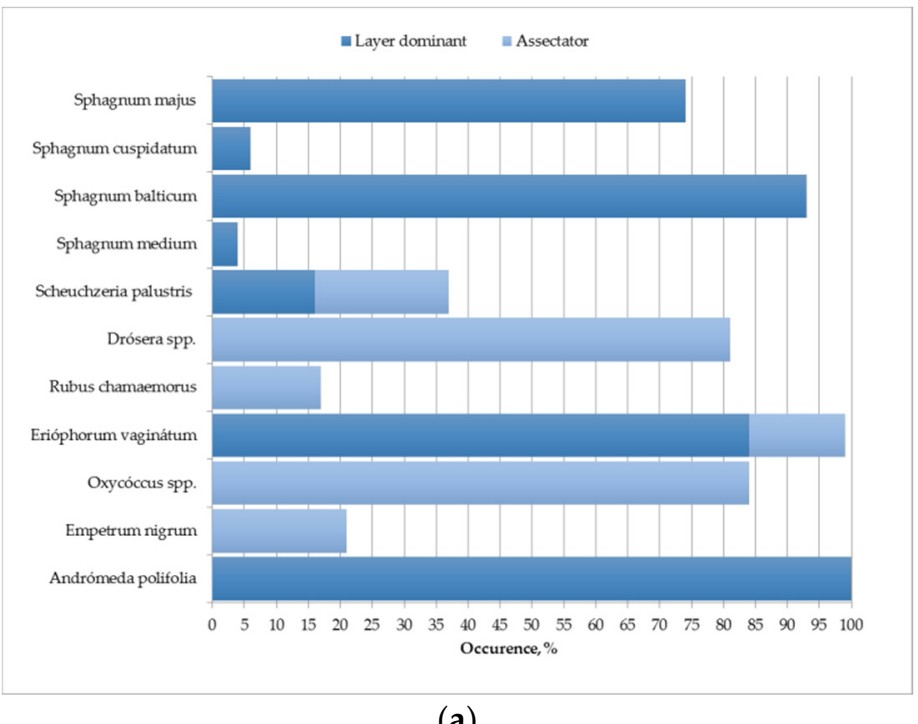

(**a**)

**Figure 5.** *Cont.*

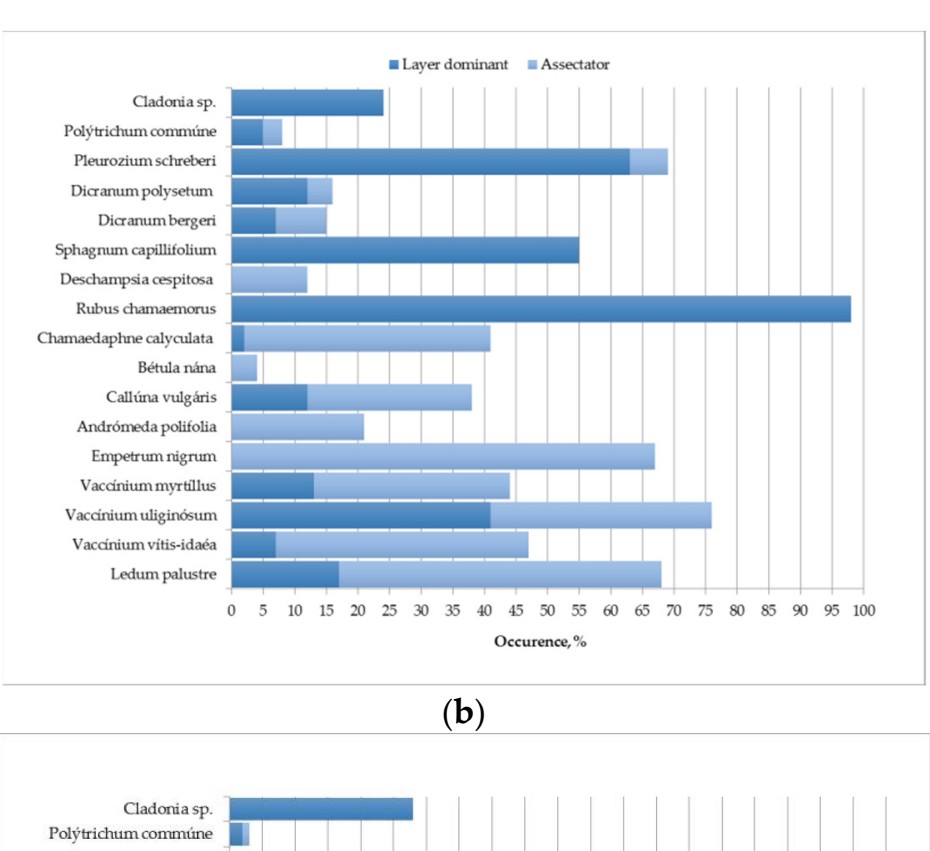

**(b)**

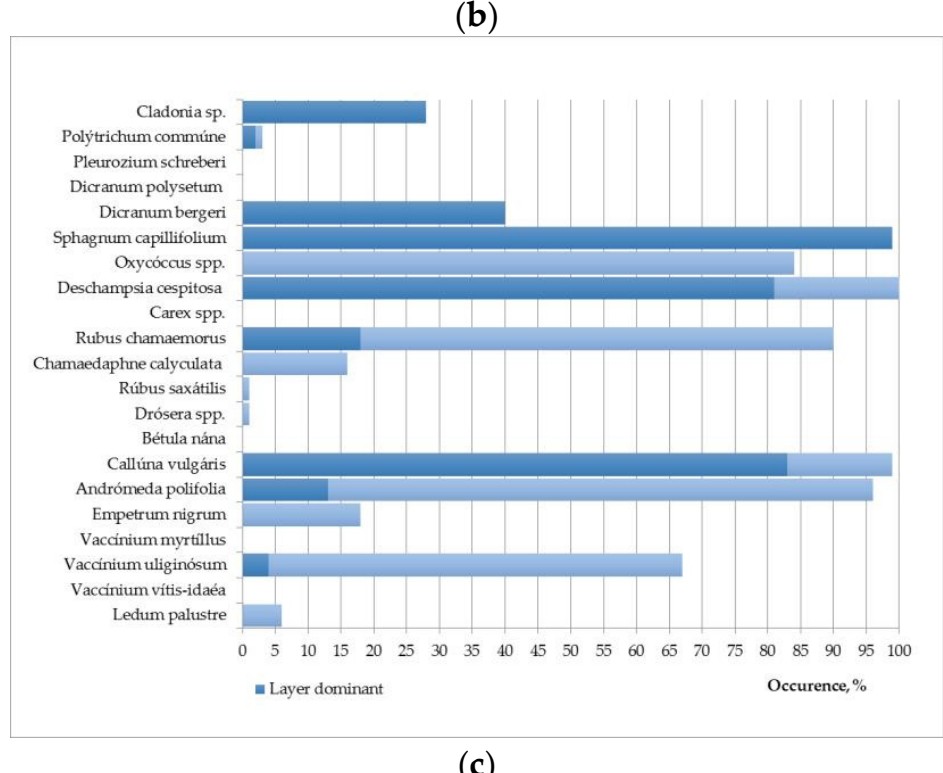

**(c)**

**Figure 5.** Average species diversity and occurrence of certain species of ground vegetation cover for: (**a**) undrained test sites; (**b**) effectively drained test sites; (**c**) rewetted test sites.

On the effectively drained test site (D-1), the thickness of the peat deposit is from 80 to 150 cm. The upper layer of the deposit (0–35 cm) is composed of high-moor, undecomposed sphagnum peat (R = 2–5%). The 35–75 cm layer is composed of cotton grass–sphagnum high-moor peat with a decomposition rate of R = 15–20%. The bottom layer (75–150 cm) is formed by pine-cotton grass peat with a degree of decomposition R = 25–30%. The peat deposit is dense. The groundwater level during the growing season varied within 15–30 cm.

In the area with secondary waterlogging (RW-1), the peat deposit has a uniform thickness of 120–135 cm. The deposit is composed of a thin (about 10 cm) upper layer of

sphagnum peat with a decomposition rate of R = 15–20%. This is followed by a layer of sphagnum peat (10–70 cm), characterized by a decrease in the degree of decomposition to R = 5–10%. The bottom layer (70–130 cm) is represented by pine-cotton grass peat with a degree of decomposition R = 20–25%, with inclusions of *Sphagnum* and *Sheuchzeria*. The peat deposit is compacted in the bottom layer. The groundwater level during the growing season varied within 0–15 cm.

### 3.2.2. Group Chemical Composition of Peat Organic Matter

Based on the data obtained on the degree of decomposition and botanical composition, the peat deposit was divided into layers. Samples for the study of the group chemical composition were taken in the most representative site of the study area. Mean values of the group chemical composition of peat samples are presented in Table 2.

The content of ash elements in peat samples from an undisturbed site is 0.8–1.9% and consistently increases with the depth. For peat samples from a drained area, it is slightly higher (1.3–2.6%) with a tendency for the distribution of ash elements along the profile, similar to the undisturbed area. In the area with secondary waterlogging, the ash content is 4.1% in the upper layer (0–10 cm), while lower along the profile; the content of ash components does not exceed 1.1–1.4%.

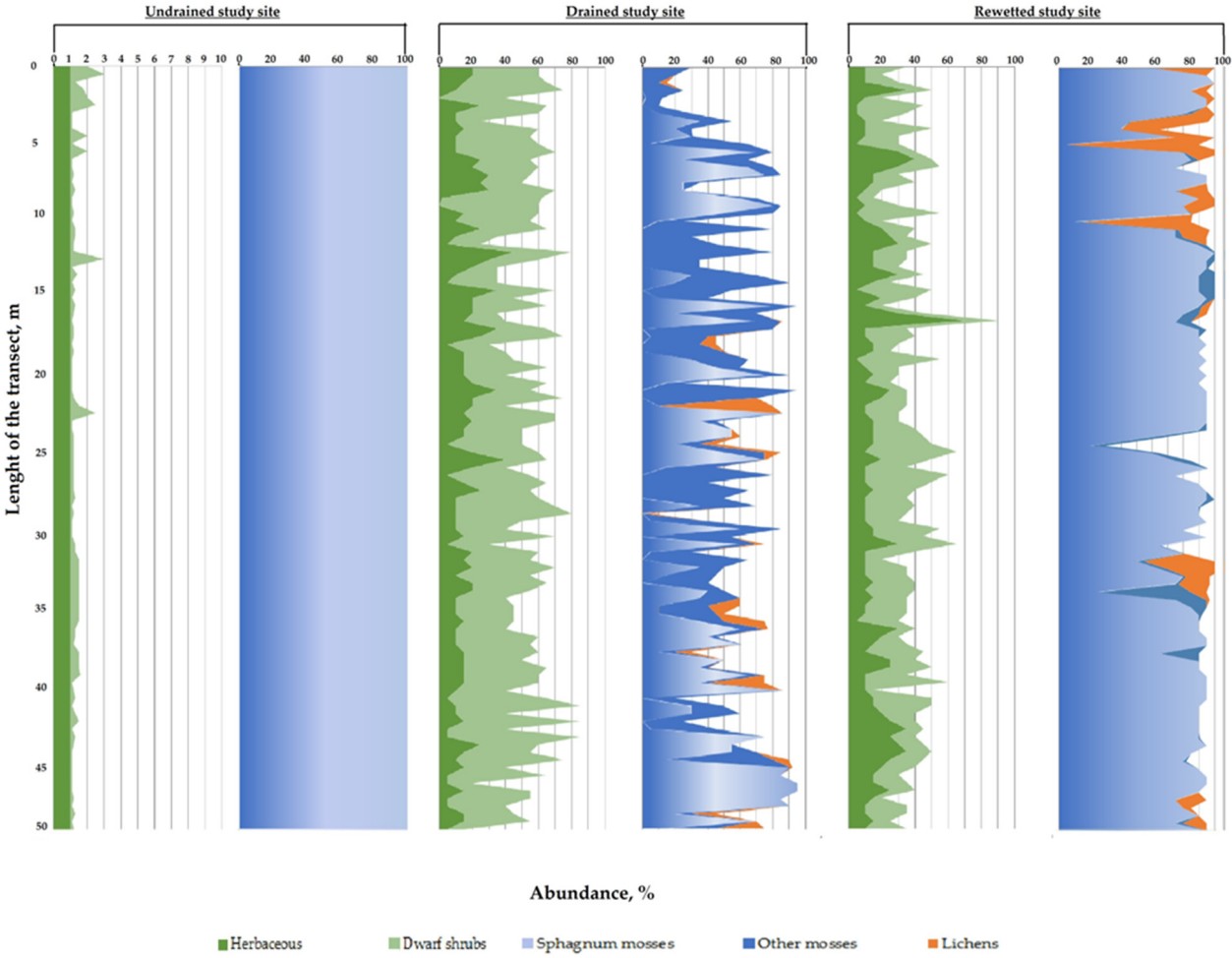

**Figure 6.** *Cont.*

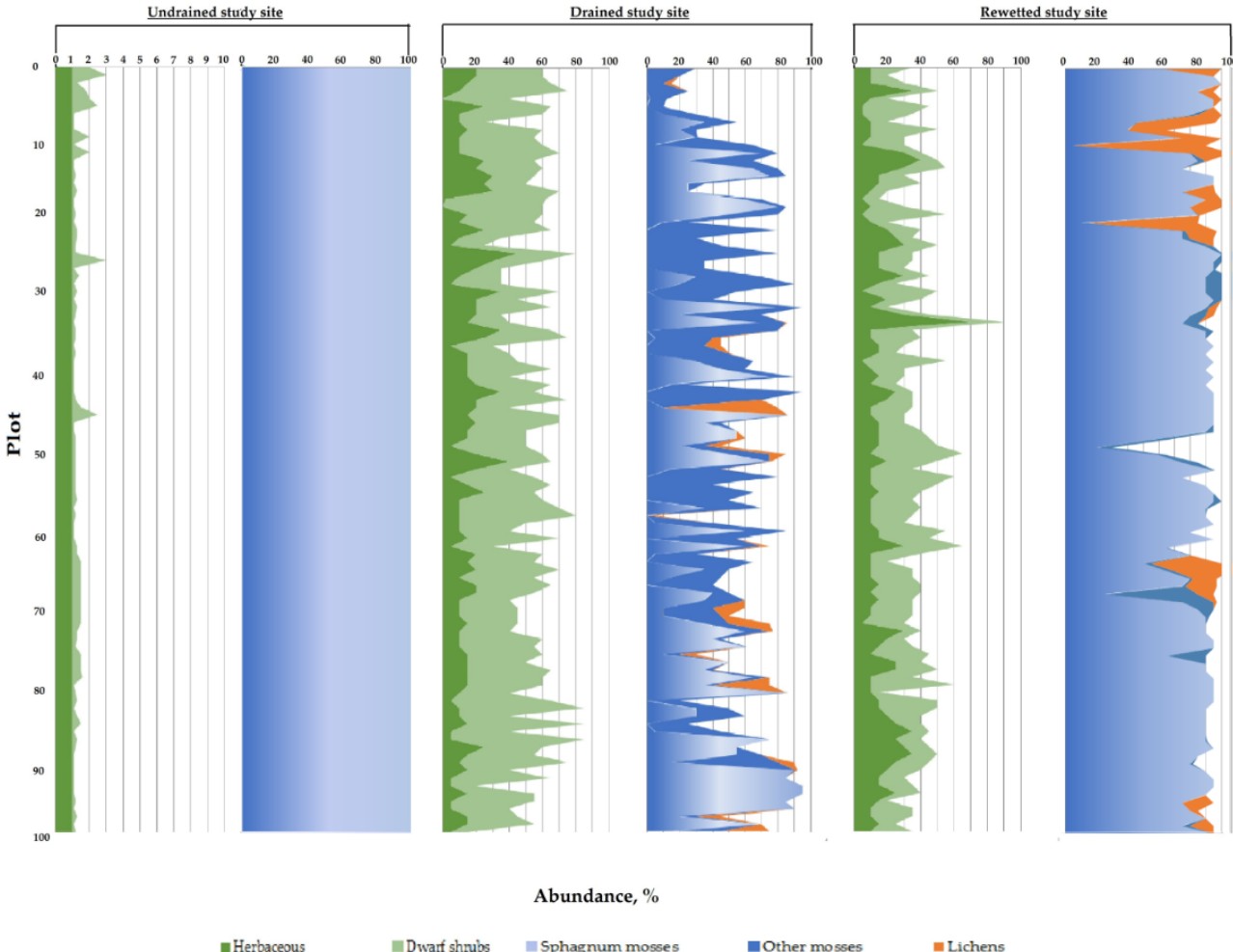

**Figure 6.** Abundance of groups of ground vegetation cover on the study sites with various degree of drainage.

The content of the lipid part in the peat samples from the undrained site is 3.9–6.0% and increases uniformly with the depth. For the drained area, the lipids' content in the lower layers (35–75 and 75–150 cm) is 8.9% and 11.1%, respectively. In the surface layer (0–35 cm), the lipids' content is similar to the samples from the undisturbed site (4.3%). In the rewetted site, a rather high content of lipid components is observed both in the upper layer (0–10 cm)—5.3%—and in the bottom layer (70–130 cm)—13.2%—a tendency for a gradual increase in the content of lipids by the depth of the deposit identical in all sites studied.

The content of water-soluble components of organic matter in peat samples is approximately the same regardless of anthropogenic impact and layer depth—1.1–1.7%.

The content of humic acids (HA) in all peat profiles increases with the depth of the deposit. On an undrained site, the HA content is 13.2–29.1%; on a drained test site—12.2–40.6%; and on the rewetted test site—13.2–31.0%. The content of fulvic acids in the peat profile of the undisturbed site decreases uniformly with depth from 8.1% to 4.2%. A similar trend is observed in the rewetted site—from 8.3% to 5.8%. In the peat profile of the drained site, the content of fulvic acids along with the profile practically does not change and remains at the level of 3.5–4.7%.

**Table 2.** Characteristics of peat deposits of the investigated study sites.

| Test Site | Depth (cm) | Degree of Decomposition R, (%) | Bulk Density (g/cm³) | Content (%) | | | | | | | |
|---|---|---|---|---|---|---|---|---|---|---|---|
| | | | | Ash | B | WSC | HA | FA | EHC | C | NHR |
| Undrained (IOB) | 5–80 | 2–5 (H1) | 0.18 ± 0.02 | 0.8 ± 0.05 | 3.9 ± 0.02 | 1.2 ± 0.06 | 13.2 ± 0.03 | 8.1 ± 0.05 | 46.2 ± 0.02 | 14.1 ± 0.10 | 14.1 ± 0.08 |
| | 80–225 | 5–10 (H1) | 0.17 ± 0.01 | 0.9 ± 0.05 | 4.2 ± 0.02 | 1.4 ± 0.06 | 15.6 ± 0.03 | 5.7 ± 0.05 | 45.6 ± 0.02 | 13.3 ± 0.10 | 15.9 ± 0.08 |
| | 225–275 | 20–25 (H3) | 0.33 ± 0.02 | 1.9 ± 0.05 | 6.0 ± 0.17 | 1.8 ± 0.13 | 29.1 ± 0.58 | 4.2 ± 0.05 | 26.5 ± 0.11 | 6.7 ± 0.10 | 27.5 ± 0.08 |
| Drained (OB-2) | 0–35 | 2–5 (H1) | 0.11 ± 0.02 | 1.4 ± 0.05 | 4.3 ± 0.05 | 1.3 ± 0.10 | 12.2 ± 0.15 | 4.9 ± 0.05 | 47.3 ± 0.02 | 11.6 ± 0.10 | 19.7 ± 0.54 |
| | 35–75 | 15–20 (H2) | 0.36 ± 0.01 | 1.3 ± 0.05 | 8.9 ± 0.04 | 1.7 ± 0.08 | 24.3 ± 0.15 | 3.5 ± 0.05 | 31.6 ± 0.14 | 6.3 ± 0.10 | 25.3 ± 0.02 |
| | 75–150 | 25–30 (H3-H4) | 0.43 ± 0.02 | 2.6 ± 0.05 | 11.1 ± 0.26 | 1.7 ± 0.03 | 40.6 ± 0.50 | 4.7 ± 0.05 | 12.7 ± 0.04 | 2.7 ± 0.10 | 28.3 ± 0.07 |
| Rewet-ted (OB-2) | 0–10 | 15–20 (H2-H3) | 0.25 ± 0.01 | 4.1 ± 0.05 | 5.3 ± 0.05 | 1.6 ± 0.02 | 13.2 ± 0.38 | 8.3 ± 0.05 | 40.9 ± 0.15 | 8.9 ± 0.10 | 23.4 ± 0.09 |
| | 10–70 | 5–10 (H1) | 0.23 ± 0.02 | 1.1 ± 0.05 | 6.3 ± 0.05 | 1.1 ± 0.07 | 14.7 ± 0.17 | 6.6 ± 0.05 | 42.9 ± 0.05 | 11.2 ± 0.10 | 18.2 ± 0.30 |
| | 70–130 | 20–25 (H2) | 0.41 ± 0.01 | 1.4 ± 0.05 | 13.2 ± 0.51 | 1.4 ± 0.04 | 31.0 ± 0.75 | 5.8 ± 0.05 | 20.2 ± 0.14 | 3.4 ± 0.10 | 26.4 ± 0.10 |

The decomposition rate according to VonPost is shown in brackets; B—bitumen; WSC—water-soluble compounds; HA—humic acids; FA—fulvic acids; EHC—easily-hydrolysable compounds; C—cellulose; NHR—non hydrolysable residue.

The content of easily hydrolysable components in the peat deposit of the undrained site to a depth of 225 cm practically does not change and makes up almost half of the total organic matter of peat (45.6–46.2%). At a depth below 225 cm, there is a sharp decrease in the content of easily hydrolysable substances to 26.5%. On the drained site, the content of easily hydrolysable substances in the upper layer of the deposit (0–35 cm) is slightly higher than in the undisturbed site—47.3%. Additionally, there is a gradual decrease in the content of easily hydrolysable substances with depth to 12.7% in the layer 75–150 cm. On the rewetted site, the content of easily hydrolysable substances in the peat organic matter is the lowest among the 3 profiles (20.0–40.9%). An almost stable high content of easily hydrolysable substances is observed up to a depth of 70 cm—40.1–42.9%—and a sharp decrease to 20.2%—in the bottom layer.

The cellulose content in the organic matter of the peat samples of the undisturbed site is slightly higher than of the transformed sites (6.7–14.4% versus 2.7–11.6%—on the drained site and 3.4–11.2%—on the rewetted site). In general, the cellulose content with depth tends to decrease uniformly on all sites.

The content of the non-hydrolysable residue in the peat core from the undrained site increases with depth from 14.4% to 27.5%. At the same time, up to a depth of 225 cm, the content of the non-hydrolysable residue practically does not change (14.1–15.9%). This is not observed on the transformed sites. On the drained site, the content of lignin-like substances increases uniformly from 19.7% to 28.3%. On the rewetted site in the upper (0–10 cm) and lower layers (70–130 cm), the content of non-hydrolysable residue is 23.4% and 26.4%, respectively; in the intermediate layer (10–70 cm), the content of non-hydrolysable residue is slightly lower—18.2%.

## 4. Discussion

### 4.1. Vegetation

#### 4.1.1. Tree Layer

The results obtained on the properties of the forest stand on the studied areas generally correspond to the literature data for oligotrophic bogs of the northern taiga [47–53]. The forest stands of raised bogs weakly react to drainage; average annual growth increases by 1.1–2.5 times.

On the rewetted study sites, the forest stand properties are slightly lower than on the drained sites. A clear decrease in the width of annual rings in model trees in the rewetted study sites has been observed since the 1980s and more actively since 1997–1998. Rewetting on the studied areas is associated with a gradual disruption of the drainage ditches, because the maintenance of the drainage systems in the study area was not carried out. As a result, the groundwater level reaches the surface even during the summer low water period, and the canals are overgrown with sphagnum mosses.

A secondary waterlogging led to some suppression of pine stands, in particular, a decrease in radial growth, while maintaining the main vital functions of the forest stand. This is quite understandable, since in oligotrophic bogs, the process of post-drainage waterlogging takes more than 35–40 years, which allows pine stands to develop a surface root system [54]. The tree layer sharply decreases its productivity during secondary waterlogging, but it is quite probable that it could overcome the stress and return to the productivity values close to the pristine bog forest stands due to the vast hydrological tolerance of the pine.

#### 4.1.2. Ground Vegetation Cover

The poor species composition of the ground vegetation cover of undisturbed areas (6 species of vascular plants; 5 species of sphagnum mosses; 1 species of liverwort) is characteristic of the pine-cotton grass–sphagnum phytocenoses of northern taiga oligotrophic peat bogs [55]. The dominants and edificators are sphagnum mosses, forming a continuous mat. In the spatial structure of the site, *Sphagnum majus*, *S. balticum* and *S. medium* dominate.

Herbs (mainly *Eriophorum vaginatum*) and shrubs (mainly *Andromeda polifolia*) are evenly distributed but occupy insignificant areas (Figures 5a and 6).

On the drained site (Figure 5b), the plant community was replaced by a *Vaccinium uloginosum–Ledum palustre*–green moss pine forest. Compared to undisturbed areas, the species composition is more diverse, the structure of phytocenoses has become more complex due to the appearance of green mosses and lichens: 11 species of vascular species were noted; 1 species of sphagnum mosses; 5 types of green mosses; and 2–5 species of lichens of the *Cladonia* genus. The participation of sphagnum mosses in the formation of the moss–lichen layer sharply decreases relative to the undisturbed area—only *S. capillifolium* is found in micro depressions, which has wide ecological tolerance (the abundance does not exceed 3%). Green mosses abundance is 25–28% of the surveyed area. Shrubs of the *Vaccinium* genus and *Ledum palustre* dominate in the herb-dwarf shrub layer with an abundance of 29–32% (Figure 6).

The plant community in the rewetted sites is heterogeneous (Figure 5c). In the vegetation cover in these sites, both species of the undisturbed and drained area were registered. In the moss–lichen layer, *S. capillifolium* is dominant with an abundance of 80–83%, species diversity in mosses decreases; *Dicranum bergeri* and single specimens of *Politrichum commune* with insignificant abundance (up to 3%) were recorded (Figure 6). The percentage of the designed coverage of herbaceous plants remains at the level characteristic of drained sites. *Calluna vulgaris* actively participates in the structure of the phytocenosis (the abundance of shrubs is 20–24%). The herbal group is dominated by *Deschampsia caespitosa* (the abundance is 16–17%) (Figure 6).

The post-meliorative dynamics of succession processes in phytocenoses is determined not only by the type of bog, the species composition of plant communities, the intensity and duration of drainage but also by the ecological tolerance of plant species, as well as by the competitive ability of species that penetrate after drainage. A change in the species composition occurs due to a decrease in the abundance or disappearance of hygrophytic shrubs with low ecological tolerance relative to the hydrological factor and an increase in the share of green and polytrichous mosses and lichens. The peat deposit, consisting mainly of sphagnum mosses, provides long-term preservation of the hydrological regime in a close to a pristine state, and a large percentage of the participation of sphagnum mosses in the vegetation cover creates unfavourable conditions for invasive species. The peculiarities of the hydrological regime and the presence of species with wide ecological plasticity in relation to the water factor restrain the expansion of dry land species to drained areas. The same fact explains the rapid post-drainage waterlogging of areas of drained oligotrophic bogs in the absence of proper maintenance of the drainage system.

### 4.2. Peat Deposit Characteristics

### 4.2.1. Degree of Decomposition and Botanical Composition

The study of the botanical composition of peat profiles confirms the predominance of sphagnum residues throughout the peat profile. A change in the direction of development of succession and changes in the composition and structure of phytocenosis during drainage leads to a change in the nature and structure of litter entering the deposit. However, due to the relatively short period from the moment of draining, this can be traced only in the upper 5–7 cm of the peat deposit of the drained area. On the rewetted study sites, these changes were not observed, due to the attenuation of the processes of changing phytocenoses and the preservation of the predominant role of oligotrophic bog vegetation (in particular, sphagnum mosses).

A noticeably greater effect on the peat deposit is exerted by a change in the boundaries of the acrotelm (the boundaries of the aerated layer increase to 40–60 cm). This is accompanied by an increase in the degree of decomposition, a change in the structure and composition of the organic and mineral components of peat (Table 2). At the same time, the processes occurring as a result of drainage lead to a significant drawdown of the peat deposit, which is due to both physical compaction of the deposit and the intensification of

biodegradation of plant residues. This process is likely due to an increase in greenhouse gas emissions and the removal of water-soluble components from the body of a peat deposit during floods.

### 4.2.2. Group Chemical Composition of Peat Organic Matter

Based on the data in Sections 3.1.1 and 3.1.2, the authors have drawn up a principal scheme of the decomposition of peat organic matter depending on hydrological conditions and the composition of peat-forming plants in a peat deposit (Figure 7).

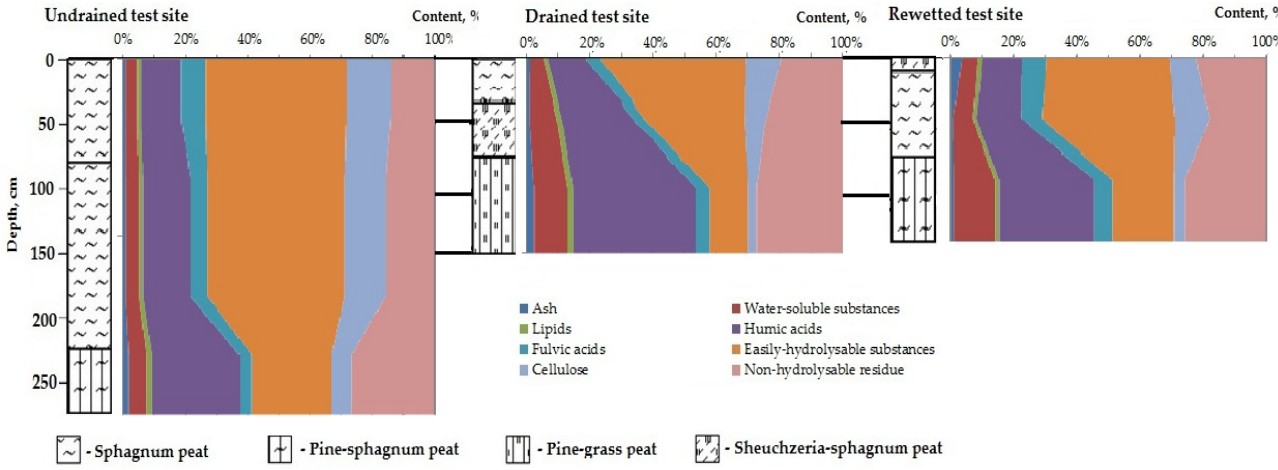

**Figure 7.** Principal scheme of the decomposition of peat organic matter.

The analysis of the results presented in Table 2 indicates that the ash content in the peat of all the studied areas varies in the range of 0.8–4.1%. This corresponds to the values obtained earlier for oligotrophic peat in bogs of the South Belomorian type [56]. Taking into account the specifics of nutrition of oligotrophic bogs, the main source of mineral components entering the body of the deposit are peat-forming plants, which in turn receive them from precipitation, i.e., due to atmospheric nutrition. As a consequence, an increase in ash content along the depth is observed for peat of undrained and drained study sites. This is due to an increase in the degree of decomposition in the bottom layer, a change in the botanical composition of peat (an increase in the proportion of herbaceous species), as well as a possible migration from the mineral bottom. The oligotrophic bogs are dependent entirely on precipitation (e.g., rainfall, snowfall) for mineral nutrition. Under natural conditions, a uniform accumulation of the mineral particles from the atmosphere occurs in the peat deposit with depth. An increase in the ash content when moving deeper into the deposit is associated with the natural destruction of plant residues processes.

The increased content of ash content in the upper 0–10 cm layer of the peat deposit of the rewetted study site is probably due to local atmospheric emissions on the study area. However, due to the greater thickness of the identified characteristic horizons in the peat deposit of the undrained study site (5–80 cm) and drained study site (0–35 cm), this increase in the ash content was not recorded.

Bituminous components (lipids, extractive substances) of peat include both labile compounds (vitamin–pigment complex) and stable ones (pentacyclic triterpenoids, steroids, n-alkanes, n-alkan-2-ones, acids of various structures and composition, etc.). It is known that the composition of extractive substances is determined by the type and conditions of plant growth. As a consequence, the content of lipids in the upper layers of the acrotelm is determined mainly by the species diversity of bioproducers (peat-forming plants), in which these substances provide the main vital functions, and by the geo-climatic conditions of biosynthesis [57–60]. Thus, an increase in the proportion of dwarf shrubs in drained study sites leads to an increase in the content of lipids in the upper layers of the acrotelm from 3.9 to 4.3 and 5.3%. The intensification of peat-destruction processes as a result of the carried



out drainage works contributes to an increase in the rate of lipids' accumulation with the depth. This is expressed in an increase in the content of lipids in the bottom layers by 1.9 and 2.2 times. It should be noted that re-waterlogging does not lead to a decrease in the content of peat lipids.

The most labile part of peat organic matter is represented by a group of low molecular weight compounds soluble in hot water. Their share in peat, as a rule, is not large and for the studied deposits varies in a narrow range from 1.1 to 1.8% (Table 2). It is believed [61] that this fraction consists of 10–20% compounds of the polysaccharide complex, and the rest consists of low molecular weight and oligomeric compounds of a phenolic nature. However, we have previously established [62] that polysaccharides are not present in analytically significant amounts in water extracts from high moor peat in the European North of Russia. On the whole, the content of water-soluble compounds in the analyzed peat samples practically does not change, as in the earlier studied high-moor peat samples of the region under study, which we studied earlier [56].

For peat of the studied peat samples, the proportion of easily-hydrolysable components varies from 12.7 to 47.3% (Table 2). In all studied sites, the maximum content of easily-hydrolysable compounds was noted in the upper layers of the acrotelm, which is due to the biosynthesis of these substances by peat-forming plants and their active participation in the maintenance of life processes of plant organisms. Naturally, the share of this compound in the acrotelm decreases with the depth due to the processes of their mineralization and condensation. At the same time, their content decreases by 2 times for undisturbed and rewetted study sites, while their share decreases more significantly—by 4 times on the drained study sites. Accordingly, the minimum values of easily-hydrolysable compounds on anthropogenically transformed study sites fall outside the lower boundary of the interval characteristic of undisturbed oligotrophic peat deposits of this natural and climatic zone [56].

It is believed that in the course of the mineralization of labile compounds (WSS and EHS), the formation of simple chemical substances ($CO_2$, $H_2O$, $CH_4$, etc.) occurs, while the products of partial decomposition are involved in the process of humification—the synthesis of new aromatic compounds (HA) [63]. The fact that, in the process of humification, the accumulation of humic acids occurs due to the condensation of the decomposition products of the easily-hydrolysable part is confirmed by a pronounced correlation between the content of HA and EHS (Figure 8). Thus, the EHS content seems to be a sensitive indicator of changes in the transformation of peat organic matter under the influence of anthropogenic load.

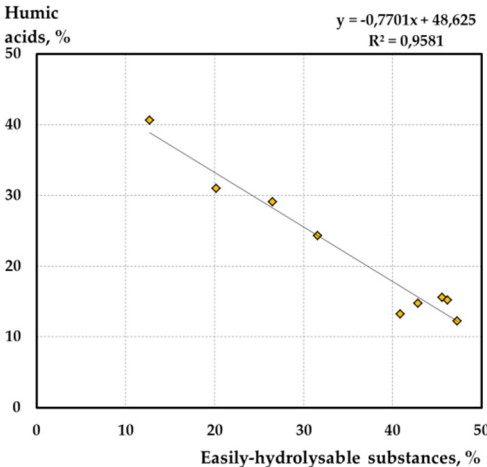

**Figure 8.** Dependence of humic acids content on easily-hydrolysable substances content of studied peat samples.

The formation of primary humic compounds occurs already at the stage of plant death. According to Lishtvan I.I., Orlov D.S., Shinkarev A.A. & Gnevashov S.G. and Zech W. et al. [63–66], in the future, they can condense with the intermediate products of destruction of plant components of aromatic and aliphatic nature, undergoing significant structural changes. The formation of a peat deposit is a long-term process that takes up to several thousand years, and the conditions in which peat is formed are constantly changing and depend on the time of formation. Therefore, with an increase in the depth of deposit, significant changes in the content and composition of humic substances are observed (Table 2). The content of HA and FA is 12.2–40.6% and 3.5–8.3%, respectively, for the deposits under study. The depth of humification (HA/FA is the mass ratio of the content of humic and fulvic acids) makes it possible to judge the type of humus formation. The parameter HA/FA regularly increases with an increase by the depth of the deposit. The humate type of humus formation (HA/FA > 2) is characteristic of the peat of the characteristic layers of all the studied sites, except for the peat of the upper aerated layer of the undrained study site and the rewetted study site, for which the HA/FA ratio is lower, which indicates the fulvate–humate type of humus formation. The proportion of humic acids increases with depth in all studied deposits. This is consistent with the previously obtained results [56]. This may be due to both an increase in the degree of decomposition and the peculiarities of the botanical composition of peat, in particular, the presence of layers formed by the remains of mesooligotrophic or eutrophic vegetation on the bottom of the deposit [67]. The drainage leads to a significant intensification of the processes of humification of plant residues, which is expressed in an increase in the content of HA in all horizons. At the same time, secondary waterlogging naturally inhibits the processes of transformation of organic matter.

The proportion of HAs in the total content of humic substances must increase with depth. The process of oxidative condensation of phenolic compounds, which are part of the fulvic acid fraction and can act as a material for the synthesis of humic acids, proceeds most actively in the presence of atmospheric oxygen. This confirms the conclusion [68] that humus formation in the upper layer of the deposit is more likely to occur by the condensation mechanism. It should be noted that under the anaerobic conditions of the catotelm, there is a possibility of oxidative reactions due to the presence of metals with variable valences, as well as the vital processes of anaerobic microorganisms [64]. Humification under such conditions most likely proceeds by the mechanism of oxidative acid formation [69], during which there is a gradual transformation of organic matter and the "maturation" of humic acids.

The content of difficultly-hydrolysable substances and non-hydrolysable residue in the studied peat samples varies in the range of 2.7–14.1% and 14.1–28.3%, respectively. The maximum values of difficultly-hydrolysable substances, which are based on the cellulose of the cell walls of the original plants, are characteristic of the upper layers of peat; then, it decreases with the depth and reaches minimum values in the bottom layers (on drained study site, up to 2.7%, and rewetted study site, up to 3.4%). For the bottom layer of the undrained site, the content of the difficulty-hydrolysable substances is twice as high and amounts to 6.7%. At the same time, from the data presented in Table 2, it can be seen that the drainage leads to a decrease in the share of difficulty-hydrolysable substances in peat. The lignin-like compounds are characterized by the opposite dynamics. On the contrary, their mass fraction increases with depth. At the same time, it seems that the key factor determining the content of non-hydrolyzable components in peat is the botanical composition of the deposit, namely, the presence of shrubs and woody species.

Thus, there is a significant change in the group chemical composition of peat, as a result of drainage accompanied by an increase in the level of aeration of the peat deposit, intensification of the processes of oxidation and biodegradation of organic residues. In this case, the change of the qualitative composition and an increase in the solubility of peat components and, as a consequence, the removal of water-soluble compounds from the body of the deposit is possible.

*4.3. The Transformation of Oligotrophic Bog during the Process of Rewetting*

The transformation of an oligotrophic bog after prolonged drainage and during the process of secondary waterlogging, even in the absence of managed land use, manifests itself in a significant change in the species diversity of vegetation, structure and spatial distribution of the vegetation cover. Pine stand under drainage conditions with a change in the type of growing conditions increases productivity insignificantly, even taking into account the change of phytocenoses. The thickness of the peat deposit does not allow the roots to receive sufficient mineral nutrition from the underlying rocks. Secondary waterlogging leads to some suppression of the stand, but the destruction of the stand does not occur.

It was found that both the phytocenosis as a whole and its individual components are sensitive to changes in hydrological conditions. However, these changes are reversible—the bog phytocenosis begins to actively recover with the destruction and overgrowing of the drainage system. The vegetation cover reacts to drainage with an active change of phytocenosis to a forest-green moss. However, the peat deposit, consisting mainly of sphagnum mosses, ensures long-term preservation of the hydrological regime in a close to initial state, which limits the settlement of upland species. Secondary waterlogging of areas as a result of disturbances in the drainage system proceeds with the restoration of oligotrophic phytocenoses and the displacement of forest species to elevated elements of the microrelief.

At the same time, the changes occurring in the body of the peat deposit are irreversible and manifested in the compaction and depletion of the deposit, an increase in the degree of decomposition of peat and the depth of humification as well as in the change in the group composition of organic matter. Edaphotope continues to develop according to the meso- or eutrophic type during re-waterlogging. In turn, the irreversible change in the physical and chemical properties of the peat deposit limits the possibilities for the restoration of species of oligotrophic habitats in the initial state. Accordingly, the volume and variety of ecosystem services provided by such ecosystems will differ significantly from those provided by intact ones.

## 5. Conclusions

The results of the conducted studies demonstrate that re-waterlogging of oligotrophic sphagnum bogs leads to an active restoration of the oligotrophic phytocenosis and the displacement of upland species characteristic of drained areas to the elevated elements of the microrelief. At the same time, the most tolerant upland species are preserved in the vegetation cover and the most tolerant bog species recover.

The course of these processes is associated with the following factors:

(1) Changes in the hydrological regime, both as a result of drainage and secondary waterlogging, cause significant changes in the species diversity of vegetation, the structure and spatial distribution of the vegetation cover. They are manifested in the expansion of upland species upon drainage, and in the emergence of the most tolerant oligotrophic species, in particular, some of the sphagnums, to dominate positions upon secondary waterlogging.

(2) Changes in the structure and chemical composition of peat deposits resulting from the drainage of the bog are irreversible and not compensated for by rewetting. This limits the possibilities of restoring the original oligotrophic habitats.

(3) Accordingly, the amount and variety of ecosystem services provided by such restored wetlands will differ significantly from those provided by intact ones.

**Author Contributions:** I.Z. and T.P. conceived and designed the experiments; A.S., O.Y. and I.Z. performed the experiments; O.Y. and S.S. analyzed the data; T.P., writing—original draft preparation; S.S., writing—review and editing the paper. All authors have read and agreed to the published version of the manuscript.

**Funding:** This research was funded by Ministry of Science and Higher Education of the Russian Federation project number № AAAA-A18-118012390224-1 and Russian Foundation for Basic Research grant number 18-05-60151.

**Institutional Review Board Statement:** Not applicable.

**Informed Consent Statement:** Not applicable.

**Conflicts of Interest:** The authors declare no conflict of interest. The funders had no role in the design of the study; in the collection, analyses, or interpretation of data; in the writing of the manuscript, or in the decision to publish the results.

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
