# Peer review of "Transformation of an Oligotrophic Sphagnum Bog during the Process of Rewetting"

_land, doi:10.3390/land10070670_

Round 1
Reviewer 1 Report
The manuscript titled “ Transformation of an Oligotrophic Sphagnum Bog after Prolonged Drainage and during the Process of Rewetting in the Absence of Land Use. “ I find the idea interesting and in line with the aim of the journal. I have some concerns about the experimental setup to justify what the authors claim. Moreover, the rationale behind some of the data presented was not entirely clear. I also recommend to the authors improve their references by conducting a more extensive review of international literature. Particularly, the introduction statements are not supported by the references selected by the authors. The logic of some sentences is also questionable. Below is my point-to-point analysis of the manuscript.
- I suggest modifying the title should be more crisp and brief.
- Abstract introductory statement is too long; it has to be improved with a more specific rationale of the study. The abstract should have crisp information about aim materials method result and conclusion, which I don't find in the present form of the abstract.
- Keywords: Please try to restrict the words to one or two for the first three Keywords.
- The introduction of manuscript is with fewer references. I recommend adding reference to the introduction.
Materials and methods
- Could you please throw more light on the procedure employed?
- Results and discussion:
- Although this section looks okay, I suggest comparing it with more studies of similar nature.
- Conclusion: This section should be brief with only star points.
Line 41-42…..20th century….Use superscript in case of “th.”
Line 107…..Revise “falls in July-August - 70-73 mm”.
In the Laboratory analysis, the estimation of moisture content, ash content, bulk density, etc., may be elaborated. Did you use a core sampler for estimating bulk density? If yes, provide the dimension (height, diameter, volume) of the core sampler.
Figure 4: Bring the x-axis title to the middle of the graph. Put the legend inside the graph. The authors have used two colours, i.e., dark and light blue. But provided only one legend.
Figure 5: Bring the x-axis title to the middle of the graph and legend to the top middle.
Table 1: Use uppercase in the first letter of age, height, diameter. Place the unit in the bracket.
Elaborate the abbreviation in the footnotes, wherever applicable.
Table 2: Place the units in brackets.
Line 447……..According to ?? Write the author's name.
Specific outcomes must be discussed in the conclusion section.
Please follow the MDPI style in the reference section.
Author Response
Dear Reviewer,
We express our gratitude to you for your careful consideration of our study and constructive comments on the prepared article. We have made the necessary amendments to the text. We will also take into account the comments of the reviewer in subsequent scientific work. please see the attachment
Kind Regards,
Tamara Ponomareva and co-authors

Reviewer 2 Report
The manuscript is well written. The contents are original and it will benefit for practical application. Some revisions are recommended prior to acceptance of the manuscript:
1. Provide additional review on recent land cover to strengthen the critical review on the changes of land cover.
2. What is justification of the selection for:
"extractives by treatment with ethoxyethane in 161 a Soxhlet apparatus; biopolymers of a humic nature by treatment with 0.1 N. sodium 162 hydroxide solution; easily hydrolysable compounds by treatment with 5% hydrochloric 163 acid solution; then the content of Klason lignin and difficult hydrolysable substances 164 were determined in the residue."
3. Why need two software in Lines 168-173? SPSS is capable to do all statistical analyses.
4. What is the implication of different ages in the study as shown in Table 1?
5. How to determine that he drainage of the studied objects was carried out 48-50 years ago (Line 189)?
6. Please provide explanation on:
"Stands of drained sites showed weak positive dynamics for drainage: the average diameter of the stand in creased from 4.1-5.6 cm to 4.5-7.4 cm, and the average height increased from 2.4-3.8 m 193 to 4.3- 8.5 m." how this measurement is done. Is it throughout the week, month or year?
7. Line 205-206: How do you know is this significant?
8. Provide reference for standard value (Lines 202-204)? Which standard is refering to?
9. Where are discussions for figure 4 and 5?
Author Response
Dear Reviewer,
We express our gratitude to you for your careful consideration of our study and constructive comments on the prepared article. We have made the necessary amendments to the text. We will also take into account the comments of the reviewer in subsequent scientific work. Please see the attachment.
Kind Regards,
Tamara Ponomareva and co-authors

Reviewer 3 Report
This paper demonstrates how northern bogs respond to secondary waterlogging following abandonment of drainage schemes, both in relation to the vegetation present and the peat structure and composition. The results show that changes within the peat structure are not reversed as the peat rewets, while vegetation cover in the rewetted areas seems to be midway between the undrained and drained sites.
My understanding of the survey design is that it is a space for time substitution, whereby sites that have never been drained act as the control, while sites with ongoing drainage act as the comparison for how the rewetted site would be if the drains hadn't naturally blocked. There are a few points in the results where this distinction could be made more clearly as technically I don't think this study monitored change over time within the rewetted site and sometimes it reads as if this is the case.
I think that this paper offers an interesting contribution to the literature on the effects of rewetting on bogs, particularly the angle of natural rewetting and the comparison of the effects on vegetation and soils. Although the results are not unexpected and completely novel I think that they are still a useful addition to the literature. The use of English language could do with a check, there are a few words / phrases that are not in common use and could be substituted for others that would make the paper easier to follow. I have picked some out in my specific comments below but the list is not exhaustive.
Specific comments I have on the manuscript are:
L46: Phytocenoses? would vegetation mean the same thing and be a more commonly understood word?
L58: presumably changes are reversible or irreversible, not both.
L59: as above remove "both"
L71-74: I'm surprised the value is that low for drained peatlands. Is there are more recent reference that could be used, especially given the ongoing drainage of peatlands in SE Asia?
L88: land use change or managed land use might be clearer
L87-89: I'm not sure I follow this sentence. Do you mean that drainage impacts on vegetation can be reversed if the species present can adapt to differing conditions?
Figure 1: a zoomed out map showing where in Russia your study site is would assist an international readership. Also, check the abbreviations in the figure legend and in the text, they seemed mismatched in places. The information on where the pristine, drained and rewetted sites are could be added to the figure legend to make it clearer.
L98: superscript km2
L104: change comma to decimal point
L135: I'm not sure what "facies" means. I'd change this word.
157-166: I struggled to follow this paragraph, could the methods be written more clearly? I also think it should be 0.1 M (not N)?
Table 1: Mensuration isn't a word I've come across before, maybe measured characteristics would be more easily understood? Also, what does normality mean in the context of the table? Why is the birch age NA? Is this natural regeneration and the other species are plantation?
L189: Studied sites (not objects)
L189-195: did forest cover increase following drainage? Also, are the comparisons between the drained and undrained sites, and if so would they originally have had the same vegetation?
L200 (and others): What do you mean by projective? I feel there will be a more widely used English word that could be used.
Figure 4: What is the reason for the light and dark blue bars in the figure? I think this would be a clearer figure if the same plant species were on the Y axis of all 3 graphs so you could see shifts in species occurrence between the areas. Alternatively, could all three graphs be displayed in 1 graph as a grouped bar chart.
Figure 5: Are these transects perpendicular to the ditch lines? If so mark where the ditches are (if present).
L316-319: Not clear what you mean here, I thought the trees grew better in the drained sites?
L380 and others: I would say the more common spelling is acrotelm (and catotelm).
Author Response

(The authors gave the same response as above.)

Reviewer 4 Report
Nice study - especially when very little on these kind of experiments have been published from Russia. There are some things that could/should be improved.
The figures in my version have some peculiarities. Fig. 5 c is other way round. Fig. 6 is missing and Fig. 7 is again named as Fig. 5. On lines 452-453 you mysteriously mention air pollution. This should be opened more. On lines 557-586 you have conclusions -type material. This should be combined as shorter with conclusions.
Author Response
Dear Reviewer,
We express our gratitude to you for your careful consideration of our study and constructive comments on the prepared article. We have made the necessary amendments to the text. We will also take into account the comments of the reviewer in subsequent scientific work.
Page Number |
Line number |
Author’s comment |
10 |
Figure 5 |
The authors agree with the comment of the reviewer. Figure 5с was corrected. |
11 |
Figure 6 |
The authors agree with the comment of the reviewer. Figure 6 was corrected. |
17 |
Figure 7 |
The authors agree with the comment of the reviewer. Figure 7 was corrected. |
18 |
452-453 |
The authors agree with the comment of the reviewer. The appropriate amendments have been made to the text of the article. |
21 |
557-586 |
The authors believe that this text should be left for the explanation of the essence of the processes taking place in general in the oligotrophic bog ecosystem during secondary waterlogging. We have added the corresponding subparagraph in the text of the paper. The appropriate amendments have been made to the text of the article. |
Kind Regards,
Tamara Ponomareva and co-authors

Round 2
Reviewer 1 Report
I appreciate the author for a nice piece of work. All my comments are successfully replied and corrections are made by the author in the manuscript. I recommend accepting the manuscript in the present form, no further revisions are required.
Author Response
Dear Reviewer,
We thank you for your great work and patience. We are very glad that you appreciated our scientific work.
Kind Regards,
Tamara Ponomareva and co-authors

Reviewer 2 Report
The Authors has addressed the comments from Reviewers properly and they have revised the manuscript siginficantly. The manuscript can be accepted now.
Author Response

(The authors gave the same response as above.)
